# Antibiotic Cocktail Effects on Intestinal Microbial Community, Barrier Function, and Immune Function in Early Broiler Chickens

**DOI:** 10.3390/antibiotics13050413

**Published:** 2024-04-30

**Authors:** Waseem Abbas, Ruichen Bi, Muhammad Dilshad Hussain, Alia Tajdar, Fangshen Guo, Yuming Guo, Zhong Wang

**Affiliations:** 1State Key Laboratory of Animal Nutrition, College of Animal Science and Technology, China Agricultural University, Beijing 100093, China; waseemabbas@cau.edu.cn (W.A.); s20223040717@cau.edu.cn (R.B.); b20203040314@cau.edu.cn (F.G.); guoyum@cau.edu.cn (Y.G.); 2MARA-Key Laboratory of Surveillance and Management for Plant Quarantine Pests, College of Plant Protection, China Agricultural University, Beijing 100193, China; dilshad@gzu.edu.cn; 3Key Laboratory of Agricultural Microbiology, College of Agriculture, Guizhou University, Guiyang 550025, China; 4Key Laboratory of Insect Behavior and Harmless Management, College of Plant Protection, China Agricultural University, Beijing 100193, China; aliatajdar@cau.edu.cn

**Keywords:** antibiotic cocktail, early broiler chickens, gut microbiota, intestinal barrier functions, immunity

## Abstract

This study investigated the effects of an antibiotic cocktail on intestinal microbial composition, mechanical barrier structure, and immune functions in early broilers. One-day-old healthy male broiler chicks were treated with a broad-spectrum antibiotic cocktail (ABX; neomycin, ampicillin, metronidazole, vancomycin, and kanamycin, 0.5 g/L each) or not in drinking water for 7 and 14 days, respectively. Sequencing of 16S rRNA revealed that ABX treatment significantly reduced relative Firmicutes, unclassified Lachnospiraceae, unclassified Oscillospiraceae, *Ruminococcus torques*, and unclassified Ruminococcaceae abundance in the cecum and relative Firmicutes, *Lactobacillus* and *Baccillus* abundance in the ileum, but significantly increased richness (Chao and ACE indices) and relative *Enterococcus* abundance in the ileum and cecum along with relatively enriched Bacteroidetes, Proteobacteria, Cyanobacteria, and *Enterococcus* levels in the ileum following ABX treatment for 14 days. ABX treatment for 14 days also significantly decreased intestinal weight and length, along with villus height (VH) and crypt depth (CD) of the small intestine, and remarkably increased serum *LPS*, *TNF-α*, *IFN-γ*, and *IgG* levels, as well as intestinal mucosa *DAO* and *MPO* activity. Moreover, prolonged use of ABX significantly downregulated *occludin*, *ZO-1*, and *mucin 2* gene expression, along with goblet cell numbers in the ileum. Additionally, chickens given ABX for 14 days had lower acetic acid, butyric acid, and isobutyric acid content in the cecum than the chickens treated with ABX for 7 days and untreated chickens. Spearman correlation analysis found that those decreased potential beneficial bacteria were positively correlated with gut health-related indices, while those increased potential pathogenic strains were positively correlated with gut inflammation and gut injury-related parameters. Taken together, prolonged ABX application increased antibiotic-resistant species abundance, induced gut microbiota dysbiosis, delayed intestinal morphological development, disrupted intestinal barrier function, and perturbed immune response in early chickens. This study provides a reliable lower-bacteria chicken model for further investigation of the function of certain beneficial bacteria in the gut by fecal microbiota transplantation into germ-free or antibiotic-treated chickens.

## 1. Introduction

To sustain a healthy lifespan, gut microbes play a crucial role in maintaining metabolic and immune homeostasis and protecting against pathogens [1,2,3]. The microbial community’s symbiosis with the host helps to maintain homeostasis and regulate immune responses [4]. However, microbial dysbiosis may resulting in dysregulation of bodily function and diseases. In chickens, a complex intestinal homeostasis is regulated by gut microbiota that involves the simultaneous morphological and immunological development of intestinal tissues [5]. In addition, antibiotic growth promoters have been administered as a feed additive for decades to improve food digestion and animal health while simultaneously helping to control microbial diseases [6]. Antibiotics in poultry feed have been demonstrated to boost feed efficiency and growth performance and minimize the levels of enteric bacterial infections, including *Clostridium perfringens* [7], *Salmonella enterica* [8], *Escherichia coli* and *Staphylococcus aureus* [9]. However, long-term and indiscriminate use of AGPs or antibiotics has some detrimental health effects and bacterial modifications that contribute to antimicrobial resistance and may extend to commensal and pathogenic bacterial microbes [10,11]. In addition, antibiotic cocktails disrupt tight junction proteins such as claudin, occludin, and zonula occludens (*ZO*), which maintain the gut barrier in mice. This can lead to increased intestinal permeability, also known as “leaky gut.” [12]. In another study, antibiotic cocktail supplementation in mice negatively impacted the gut microbiota and immune-related genes by downregulation of signaling pathways, which affects innate lymphoid cell 3 (*ILC*3) at an early age in mice and leads to inflammation [13]. A dietary supplementation of AGPs substantially reduces the abundance of *Clostridium perfringens* and other Gram-positive bacteria, such as *Lactobacilli*, *Bifidobacteria*, and *Streptococcus*, which make up the majority of beneficial bacteria in the gastrointestinal tract [14], while increasing the proliferation of Gram-negative bacteria including *Salmonella* and *Campylobacter*, possibly due to a lack of competition for available nutrients [15]. An antibiotic cocktail containing ampicillin, vancomycin, neomycin, and metronidazole that induces alteration in the microbiome and metabolism has been extensively employed [16]. In a mouse model, antibiotic cocktail administration significantly alters the gut microbiota composition (such as eliminating microbes belonging to Actinobacteria, Bacteroidetes, and Verrucomicrobia), inflammatory responses, and short-chain fatty acids (SCFAs) in the cecum and shows negative effects on intestinal maturation [13,17,18]. However, the influence of early-life prophylactic antibiotic cocktail applications on the gut microbiota and disease resistance requires further investigation in broilers. This current study aimed to evaluate the effect of the duration of treatment via drinking water with antibiotic cocktails (containing neomycin, ampicillin, metronidazole, vancomycin, and kanamycin) on broiler gut microbiota composition and immune function.

## 2. Results

### 2.1. Effects of Antibiotic Cocktail Treatment on Intestinal Length and Weight and Internal Organ Index

As shown in Table 1, compared with the CC group, the AC and BC broilers treated with ABX for 7 and 14 days, respectively, showed almost no significant differences in average body weight of 7-day-olds or 14-day-olds. Furthermore, the organ index (liver, spleen, and bursa) of AC and BC did not show any significant difference from the control group on day 14 either. However, duodenum length (*p* = 0.001) and weight (*p* < 0.001), ileum weight (*p* < 0.001), and total small intestine length and weight (*p* < 0.001) were reduced significantly by ABX treatment for 7 or 14 days. Furthermore, a notable reduction in jejunum weight and total small intestine weight was observed in chickens after ABX treatment for 14 days rather than for 7 days, but there was no significant difference in jejunum or ileum length at 14 days of age.

### 2.2. Effects of Antibiotic Cocktail Treatment Duration on Intestinal Morphological Structure

Compared with the CC group chickens, at day 14, villus height (VH) (*p* < 0.001) and VH/CD ratio in the duodenum, jejunum, and ileum had decreased significantly, while crypt depth (CD) had increased markedly in the jejunum after ABX treatment for 7 days or 14 days (Table 2). Furthermore, chickens fed with ABX for 14 days had lower (*p* < 0.001) villus height (VH) in different gut segments, higher (*p* < 0.001) ileum CD and reduced VH/CD ratio (*p* < 0.001) in the jejunum and ileum than chickens fed with ABX for 7 days.

### 2.3. Effects of Antibiotic Cocktail on Cecal Short-Chain Fatty Acids

On day 14, the effect on fatty acid profiles in broilers treated with ABX for 7 days and 14 days were assessed (Table 3). The results showed that concentrations of acetic acid, butyric acid, and isobutyric acid in the cecum digesta of the BC group were reduced significantly (*p* < 0.001) compared to the control CC group and the AC group (treatment with ABX for 7 days). Moreover, the BC group chickens had lower acetic acid, propanoic acid, butyric acid, isobutyric acid, and isovaleric acid content in the cecum than the chickens treated with ABX for 7 days (AC). Surprisingly, the AC group chickens showed higher levels (*p* < 0.001) of acetic acid, propanoic acid, and butyric acid in the cecum than the control CC group.

### 2.4. Effects of Antibiotic Cocktail on Immune Functions

As shown in Table 4, compared to the control CC group, ABX treatment for 14 days notably increased TNF-α (*p* = 0.001), IFN-γ (*p* = 0.033) and IgG levels and reduced (*p* < 0.01) IgA content in the serum, but the AC group chickens showed increased (*p* = 0.011) IgG and reduced (*p* < 0.01) IgA levels in the serum. In addition, higher levels of TNF-α were observed in the BC group (*p* < 0.001) on day 14 compared with the AC group. Surprisingly, ABX treatment for either 7 or 14 days markedly downregulated intestinal TNF-α mRNA levels compared with the CC group (*p* = 0.016). Moreover, ileal *IL-1β*, *IL-8* and *IFN-r* mRNA abundance was notably reduced by ABX treatment for 14 days compared with the control group, but there were no significant differences in the ABX group treated for 7 days.

### 2.5. Effects of Antibiotic Cocktail on Ileal Permeability and Barrier-Related Gene Expression

As shown in Table 5, on day 14, serum lipopolysaccharide concentration had increased significantly (*p* = 0.002) in the ABX-treated groups (AC and BC group) compared to the CC group. DAO activity in the ileum mucosa in the BC group on day 14 was higher (*p* = 0.006) than in the AC group and the CC group. The BC group had higher MPO activity than the CC group and showed an increased trend for MPO activity relative to the AC group.

The RT-qPCR results showed that the antibiotic cocktail treatment for 7 days or 14 days in drinking water both significantly downregulated occludin, ZO-1 and mucin 2 gene expression on day 14 (*p* < 0.05) compared to the control group. Meanwhile, the mRNA levels of claudin 1 had reduced significantly (*p* < 0.05) after 14 days of ABX treatment, but there was no significant difference observed in the AC group compared with the CC control. No significant influence on FABP-2 expression in the ileum was observed after 7 days or 14 days of antibiotic cocktail application (*p* = 0.558) (Table 5).

As illustrated in Figure 1, chickens given ABX for 7 days showed significantly reduced ileal GC numbers at 14 days, and ileal GC counts successively reduced with prolonged treatment with ABX for 14 days.

### 2.6. Effects of Antibiotic Cocktail on Cecal Microbial Composition

On day 7, significant differences in operational taxonomic units (OUTs), α diversity, β diversity, and composition of gut microbiota were observed between the control and the ABX-treated groups (Appendix A).

On day 14, a total of 4328 OTUs were obtained from ceca contents of the three groups based on 97% sequence similarity. Among them, 195 common core OTUs were shared by the three groups, while 1011, 868, and 2449 OTUs were unique to groups CC14, AC14, and BC14, respectively (Figure 2A). The α diversity was significantly influenced by ABX treatment for both 7 days and 14 days. Chao and ACE indices were significantly increased, whereas Shannon and Simpson indices were not affected significantly in the cecum of the ABX-treated chickens on day 14 (AC 14 and BC14) compared to the control (Figure 2B). On principal component analysis (PCoA), β diversity showed different microbial communities among all three groups in the cecum of broilers (Figure 2C). Furthermore, at the phylum level, Firmicutes shows less relative abundance in BC14 group than the CC14 and AC14 groups on day 14 (Figure 2D, Appendix A). At the genus level, relative Enterococcus abundance increased significantly in the BC14 group compared to AC14 and CC14. However, unclassified Lachnospiraceae, Oscillospiraceae, Ruminococcus torques, and unclassified Ruminococcaceae showed a notable reduction in relative abundance in the BC14 group compared to the AC14 and CC14 groups on day 14 (Figure 2E, Appendix A). LEfSe analysis showed that group BC14 had significantly enriched Enterococcus and Enterococcaceae and the AC14 group had significantly enriched Oscillospirale and Oscillospiraceae. Clostridia and Firmicutes were rich in the CC14 group (Figure 3A,B).

### 2.7. Effects of Antibiotic Cocktail on Ileal Microbial Composition

At 7 days of age, significant differences were observed in OUTs, α diversity, β diversity, and the composition of the gut microbiota between the CC control and ABX treatment groups (Appendix A). On day 14, 1365, 3566 and 1308 unique OTUs was found in the ileum from the AI, BI, and CI groups, respectively. Based on 97% sequence similarity, 164 common core OTUs were shared by the three groups (Figure 4A). The Chao index and ACE index were significantly higher in the ileum of the BI group than CI14 and AI14. However, no significant difference in Shannon or Simpson indices was observed among the groups (Figure 4B). PCoA analysis showed that the β diversity of BI14 was different from that of the other two groups, CI14 and AI14 (Figure 4C). Firmicutes, Bacteroidetes, Cyanobacteria, Actinobacteria, and Proteobacteria were dominant bacteria at phylum level. Relative Firmicutes abundance had significantly reduced and relative levels of Bacteroidetes, Proteobacteria, and Cyanobacteria had increased remarkably in the ileum of the BI14 group compared with the AI14 and CI14 groups on day 14 (Figure 4D, Appendix A). The relative abundance of the top 10 species at the class and genus level on day 14 showed Enterococcus was significantly higher in B14, while Lactobacillus and Bacillus abundance was significantly reduced relative to AI14 and CI14 (Figure 4E, Appendix A). Furthermore, the LEfSE analysis revealed that the f Lactobaccilli and g lactobacillus in CI14, o Bacilli, f Baccilli, and g Baccilli in AI14, and g Enterococcus and f Enterococceae in BI14 were more dominant (Figure 5A,B).

Spearman correlation coefficients showed that the relative abundance of Lactobacillus was significantly positively correlated with total small intestine length and weight as well as SCFAs and negatively correlated with TNF-α (*p* < 0.05). Relative Ruminococcus torques abundance revealed positive correlations (*p* < 0.05) with total small intestinal length, cecal acetic acids, butyric acid, villus height, and goblet cell numbers in the ileum. Relative unclassified Ruminococcaceae, unclassified Oscillospiraceae, and unclassified Oscillospiraceae abundance was significantly (*p* < 0.05) negatively correlated with serum LPS, IFN-γ, and TNF-α (*p* < 0.05) and positively correlated with mRNA levels of ZO-1, claudin 1, mucin 2, small intestinal weight, VH/CD, and goblet cell density in the ileum. The relative abundance of Enterococcus showed a significant negative correlation (*p* < 0.05) with small intestinal length and weight, VH, goblet numbers, VH/CD, concentrations of cecal SCFAs (acetic acid, butyric acid, isovaleric acid, and iso-butyric acid), as well as mRNA levels of mucin 2 and ZO-1. Serum LPS, IFN-γ and TNF-α displayed significant positive correlations (*p* < 0.05) with the relative abundance of Enterococcus (Figure 6).

## 3. Discussion

Antibiotics are widely used as feed supplements and anti-infective medicine in the husbandry industry to prevent disease and promote the growth of livestock and poultry. However, the impacts of the duration of antibiotic cocktails on intestinal development, and microbiome and immune function in early chickens were unclear. This study mainly investigated the impacts of an antibiotic cocktail administered in drinking water on the intestinal microbiota composition and immune function in the early broiler chickens over a period of 2 weeks.

Our data showed that the antibiotic cocktail had no effect on body weight gain or the organ index (liver, spleen, and bursa) of broiler chickens, which was consistent with a previous study from Li et al. [19], who reported that an antibiotic cocktail treatment for 5 weeks did not affect the laying rate, feed conversion efficiency, average egg weight, or egg quality of laying hens, suggesting that the antibiotic cocktail treatment had no significant effect on growth performance and did not induce observable changes in these organs of broilers, even for extended ABX treatment. Interestingly, the length and weight of the small intestine, villus height and VH/CD ratio had reduced significantly while CD increased notably in the small intestine of chickens after 7 days or 14 days of ABX treatment compared to the CC control in our study. Moreover, ABX treatment for 14 days showed a significant reduction in the organ index compared to ABX treatment for 7 days. Similarly, previous findings showing that antibiotic cocktails or antibiotic growth promoter treatment notably decreased intestinal length and weight, further impairing the morphology in chickens and mice (lower VH and VH/CD ratio) [20,21,22,23]. Therefore, our observations suggested that ABX treatment can inhibit/decrease intestinal morphological development, and these inhibitory effects on gut development were more serious with prolonged use of the antibiotic cocktail.

Serum cytokine and immunoglobulin levels are involved in systemic immune responses in broilers. LPS concentration and DAO activity in blood as well as MPO activity in the intestinal mucosa are important indices to evaluate intestinal barrier function, which can reflect the integrity and damage degree of intestinal mechanical barrier in animals and poultry. In the current study, drinking water with ABX for 14 days significantly increased serum pro-inflammatory cytokine *TNF-α* and *IFN-γ* levels and downregulated cytokine (*IL-1β*, *IL-8*, *TNF-α* and *IFN-γ*) mRNA abundance, similar to the results of previous researchers, who reported that the mRNA levels of *TLR-4*, *MYD88*, *NF-κB*, *IL-1β*, *IFN-γ*, and *IL-4* and the ratio of *IFN-γ* to *IL-4* and/or *IL-10* in the ileum of laying hens or *IFN-α*, *IFN-β* and *IL-22* in the respiratory tract (trachea and lung) of broiler chickens were significantly downregulated following antibiotic cocktail treatment in laying hens [19] or broiler chickens [23,24,25]. At the same time, ABX treatment for 7 or 14 days remarkably increased serum LPS and IgG concentration and reduced IgA content in the serum, and prolonged ABX treatment further upregulated intestinal mucosa DAO and MPO activity. Consistent with our results, ref. [24] observed that oral antibiotics (0.25 g/L ampicillin and 0.5 g/L neomycin) in drinking water enhanced antibody responses (IgM, IgA, IgG) by two- to threefold compared with an antibiotic-free control. Upregulated pro-inflammatory cytokine TNF-α and IFN-γ levels along with raised DAO activity and IgG content in serum, possibly attributable to the overstimulation and high concentration of LPS in the immune system, where blood LPS was from Gram-negative bacteria in the inflamed gut induced by ABX. Serum IgA mainly comes from the gut, in which the intestinal immune system including B cells is stimulated by intestinal commensal bacteria and gut microbiota-derived metabolites. Reduced serum *IgA* levels might due to a depletion in gut commensal bacteria or damaged gut mucosal immune system induced by the extended ABX application, because inflamed gut usually produces less *IgA* than healthy gut. Therefore, our findings indicated that prolonged administration with ABX might increase intestinal permeability and weaken intestinal mucosal immune function, resulting in systemic inflammation.

Intestinal epithelial tight junction (TJ) barrier and its associated proteins, including claudin 1, ZO-1, and occludin, which are always used to assess epithelial barrier damage, play a crucial role in regulating nutrient absorption, protecting the integrity and permeability of the intestines, resistance against the invasion of pathogens, and maintaining intestinal homeostasis and overall health [26,27], Mucin 2 plays a significant role in the protection and lubrication of the intestinal mucosal epithelium, which maintains intestinal health [28]. Beneficial bacteria in the intestines promote the secretion of mucin 2, leading to a thicker mucin layer, while harmful intestinal bacteria break down the mucin layer, which increases intestinal permeability and damages the barrier function [28,29]. *FABP-2* served as a biomarker that reduced the integrity of the intestinal barrier epithelial cells, which indicates damage to the intestinal barrier function [30]. In this study, antibiotic cocktail treatment for 7 days or 14 days in drinking water both significantly downregulated ileal *GC* cell numbers and *occludin*, *ZO-1* and *mucin 2* gene expression. Furthermore, the mRNA levels of claudin 1 reduced significantly after 14 days of drinking ABX, suggesting that extended antibiotic cocktail application induced intestinal tight junction barrier dysfunction in early chickens. In accordance with our findings, Feng et al. (2019) [12] observed that the administration of a broad-spectrum antibiotic cocktail comprising ampicillin (1 g/L), neomycin sulfate (1 g/L), metronidazole (1 g/L), and vancomycin (0.5 g/L) in drinking water significantly disrupted the intestinal tight junction barrier in mice, demonstrated by increased permeability of the intestines to FITC–dextran, a reduction in the expression of tight junction proteins, and altered morphology of ZO-1. Moreover, antibiotic exposure affected intestinal barrier function by disrupting the intestinal microbiota or other mechanisms in animals [31,32,33,34]. Systemic inflammation has close links with the integrity of the intestinal epithelial barrier and the thickness of the mucus layer. Based on our findings, we suggest that prolonged ABX exposure thinned the protective mucus layer, disrupted intestinal barrier function and increased intestinal permeability, possibly due to the deprivation or depletion (loss) of intestinal commensal microbiota or the dysregulation of gut microbiota ecology or intestinal microbiota disturbance induced by ABX [35,36], thereby either compromising intestinal mucosal immunity and predisposing to enteric infection or promoting intestinal endotoxins to enter the bloodstream, resulting in systemic inflammatory responses [37]. Antibiotics cocktail possibly disrupt intestinal epithelial barrier function and increase intestinal epithelial permeability in broiler chickens. In other words, early exposure to antibiotics or prolonged use of antibiotics in drinking water possibly further increases the risk of various diseases in broiler chickens by disturbing gut microbiota ecology, reducing intestinal mucosal immune defense function, compromising intestinal barrier function, and increasing intestinal permeability.

Different studies indicated that the intestinal microbiota plays an important role in the development of the immune system through the improvement in the intestinal epithelial barrier, optimization of nutrient absorption, and the prevention of pathogen colonization and intestinal homeostasis [38,39]. Although it is widely accepted that the use of antibiotics as growth promoters can alter the composition of gut microbiota, either by promoting or inhibiting the growth of specific microbial species [14,40], there is a notable lack of information regarding the effect of antibiotic mixtures on the richness (the number of different bacterial species) and the evenness (the distribution of these species) of microbial communities in the chicken gut. In this study, 16S rRNA sequencing revealed that ABX treatment had a notable impact on the α-diversity index (richness) of the ileal and cecal microbiome, as indicated by higher richness (Chao and ACE indices) in the ileum and cecum after drinking ABX for 7 and 14 days, which was different from previous findings in chickens [15,23,41,42], but was similar to other studies, which revealed that antibiotic supplementation had a notable effect on α-diversity index of the cecal or ileal microbiome [43,44,45]. The conflicting data can be attributed to several factors, including the types and characteristics of antibiotics used, composition of antibiotic cocktail used, dose and duration of antibiotic addition, different gut segments and sampling time points, chicken age, diets, etc. [14]. Increased ileal and cecal microbiota species richness after extended administration of ABX are possibly attributable to the emergence of antibiotic-resistant species in the ileum and cecum following prolonged use of antibiotics [14,46]. β-diversity analysis showed that ABX treatment for 7 and 14 days showed a significant difference in community composition in the ileum and cecum compared with the control, which was consistent with results in broiler chickens treated with antibiotic growth promoters [15]. Our data showed that the ABX cocktail administered to early chickens significantly altered the diversity of gut microbiota (structure) and this alteration got more pronounced as extended use of ABX.

In our study, prolonged application of ABX significantly lowered Firmicutes, *Unclassified Lachnospiraceae Unclassified Oscillospiraceae*, *Ruminococcus torques* group and *Unclassified Ruminococcaceae* relative abundance, but increased *Enterococcus* relative abundance in the cecum. Meanwhile, our study also demonstrated a significant reduction in the abundance of relative Firmicutes, *Lactobacillus* and *Baccillus* abundance and a remarkable enrichment in relative Bacteroidetes, *Proteobacteria* and *Cynobacteria* and *Enterococcus* levels in the ileum following ABX treatment for 14 days. Similarly to our observations, broiler chickens treated with monensin, monensin–virginiamycin, and monensin–tylosin exhibited a decrease in bacilli in the cecum at 7, 14, and 35 days of age [43]. In-feed antibiotics and coccidiostat complex decreased the relative abundance of *Ruminococcaceae* and *Lactobacillaceae* in the cecum of broiler chickens [15]. Likewise, a notable depletion or drop in genus *Lactobacillus* in the chicken cecal microbiome was also observed following treatment with antibiotics, such as salinomycin (SA), bacitracin methylene disalicylate (*BMD*) [47], or ampicillin and neomycin [24], bacitracin methylene disalicylate, tylosin and virginiamycin, and ionophores, including monensin and salinomycin [45] or antibiotic cocktails [23]. Furthermore, supplementation with salinomycin has been shown to increase the abundance of pathogenic bacteria, including class *Gammaproteobacteria* [48] and family *Enterobacteriaceae* [15,40,49]. In addition, the numbers of total bacteria, *Romboutsia*, *Enterococcus*, *Aeriscardovia*, and lactic acid bacteria (*Lactobacillus delbrueckii*, *Lactobacillus aviarius*, *Lactobacillus gasseri*, and *Lactobacillus agilis* in the ileal chime) were significantly reduced by antibiotic cocktail treatment in laying hens [19]. In contrast, antibiotic cocktail (1000 IU/mg, neomycin 0.5 g/kg and 1500 IU/mg ampicillin 1 g/kg) treatment elevated the relative abundance of phylum *Firmicutes*, family *Ruminococcaceae* and *Lachnospiraceae*, and introduced *Rikenellaceae* and *Enterobacteriaceae* [36]. Adewole et al. demonstrated that birds treated with BMD had higher *Oscillospirales* abundance [50]. The variable observations were possibly attributed to the differences in composition of antibiotic cocktail used, dose and duration of antibiotic addition, sampling time points, chicken age, diets, etc.

Studies have shown that certain species of *Firmicutes* are associated with the decomposition of polysaccharides and production of butyrate, while Bacteroidetes species are responsible for breaking down complex carbohydrates and synthesizing propionate through the succinate pathway [51]. The phylum Firmicutes is particularly important during the early stages of a chicken’s life for promoting intestinal cell growth and improving the animal’s energy-harvesting efficiency [52]. Ruminococcaceae, which belongs to the class Clostridia, is known for breaking down cellulose and starch and producing butyrate with anti-inflammatory properties that contribute to intestinal development, intestinal barrier protection by upregulating TJ protein expression, and improvement in feed conversion in chickens [53,54]. Unclassified *Oscillospiraceae* and *Lachnospiraceae* are important for intestinal health, productivity, and SCFA synthesis [55,56]. *Lachnospiraceae* is confirmed to be positively associated with improved growth performance and feed efficiency in birds [54]. *Oscillospirale* and *Oscillospiraceae* were positively involved in butyrate, propionate and mucus production [57] and negatively with the population of pathogenic bacteria, including *Streptococcus* [50]. *Bacteroides* is a normal intestinal flora that is important for SCFA production and involved in the degradation of complex carbohydrates and reduced intestinal inflammation, and it also could evolve into a pathogenic form and increase when the gut is pathologically changed or impaired [58]. The phylum Proteobacteria contains Gram-negative bacteria that are potentially pathogenic and have been associated with the pro-inflammatory cytokine profile in chickens [56]. The *Enterobacteriaceae* family makes up less than 1% of a healthy gut microbiome. Although its presence is critical for keeping the immune system balanced in the gut, a rise in the population of this family can lead to significant economic losses due to enteric diseases, including mucosal ulceration [49]. The phylum *Cyanobacteria* carried a large number of antibiotic resistance genes and was closely related to inflammation and aging [59]. *Enterococcus*, the main cause of enterococcosis, a secondary disease in poultry, has become a global challenge and identified as a potential biomarker of intestinal inflammation and bowel disease [60,61,62], and antibiotic-resistant strains are frequently found in the gastrointestinal tract of chickens [63,64]. Probiotic *Lactobacillus* species might promote gut defense function by competitive exclusion of intestinal pathogens [65,66], enhanced mucosal IgA antibody production or via activation and enhancement of local cell-mediated immunity [67]. Thus, our results reflected that prolonged intervention with ABX in early chickens inhibited the growth of beneficial species and encouraged the proliferation of pathogenic bacteria, especially potential pathogenic antibiotic-resistant strains, resulting in the dysbiosis of the gut microbiota. At the same time, our data also indicated that the decreased potential beneficial bacteria were positively correlated with gut health-related indices, while the increased potential pathogenic strains were positively correlated with gut inflammation and gut injury-related parameters. Overall, prolonged intervention with ABX in early chickens is harmful to intestinal health, which possibly provides a reasonable explanation for damaged gut barrier function and compromised immune functions in the gut of early chickens after feeding with ABX.

The profound influence of gut microbiota on the host is strongly associated with gut microbiota metabolites and microbiota-derived metabolites, such as SCFAs, bile acids, amino acids, vitamins, etc. SCFAs are primarily produced by bacterial fermentation in the gut as an important source of energy for enterocytes, and are vital for intestinal health by suppressing the growth of enteric pathogens [68,69], reducing intestinal inflammation, and enhancing barrier function by altering TJ formation [70,71,72,73,74]. In this study, our results showed a significant decrease in the concentrations of SCFAs, especially acetic acid and butyric acid, in the cecal contents of the chickens treated with ABX. Similarly, a previous study showed that long-term exposure to an antibiotic cocktail resulted in decreased SCFA production [75]. This may be due to the reduction or depletion in the SCFA-producing microbiota and gut microbiota dysbiosis that resulted from the early intervention with ABX [76]. Meanwhile, the relative abundance of *Enterococcus* showed a significant negative correlation with concentrations of cecal SCFAs and gut health-related indices, but a positive correlation with gut inflammation and gut injury-related parameters [77]. Therefore, we suggest that early intervention with ABX can increase the colonization of some pathogenic antibiotic-resistant bacteria by a decreased acidic intestinal environment resulting from the decreased concentrations of *SCFAs* in the cecal contents of broiler chickens. The loss of potential beneficial intestinal strains and decreased *SCFA* levels in the gut might provide a reasonable explanation for reduced VH and VH/CD, downregulated ileal GC cell numbers and *occludin*, *ZO-1* and *mucin 2* mRNA abundance, damaged intestinal barrier integrity, increased intestinal permeability and systemic inflammation, as well as compromised intestinal mucosal immune responses observed in the early chickens treated with the antibiotic cocktail.

## 4. Materials and Methods

### 4.1. Ethics Statement

All animal experiments included in this study were approved by the China Agricultural University Animal Care and Use Committee, Beijing, China (approval AW51112202-1-2).

### 4.2. Experimental Designs

A total of 90 healthy 1-day-old male broiler chicks were obtained from a commercial hatchery (Beijing Arbor Acres Poultry Breeding Company, Beijing, China). These birds were weighed and randomly assigned to three treatments. Each treatment contained 30 birds housed in an environment-controlled laboratory animal isolator. Treatment groups were as follows: a negative control group without treatment (CC); a 1-week antibiotic-treated group with antibiotic cocktail treatment in drinking water for 1 week from day 1 to day 7 (AC); and a 2-weeks consecutive antibiotic-treated group (BC) with the antibiotic cocktail adding to drinking water for 2 weeks from day 1 to day 14. The room temperature was maintained at 32–34 °C for the first three days after hatching and gradually decreased to 22–24 °C. All birds were exposed to constant light for the first 24 h before being kept on a 23 h light/1 h dark schedule for the rest of the experiment. In addition, the broiler chicks were vaccinated on the 7th day of the experiment Throughout the study, all birds were granted unrestricted access to food and water. Depending on their assigned treatments, the chickens received diets in pellet form that were either free of antibiotics or free of coccidiostats. These basal diets were designed to meet or surpass the nutritional requirements set forth by the National Research Council in 1994. Details on the composition and nutrient content of the basal diet are given in Appendix A.

### 4.3. Antibiotic Cocktail Preparation

An antibiotic cocktail containing neomycin 0.5 g/L (N6386, Sigma-Aldrich, Saint Louis, MO, USA), ampicillin 0.5 g/L (A5354, Sigma-Aldrich, Saint Louis, MO, USA), metronidazole 0.5 g/L (M3761, Sigma-Aldrich, Saint Louis, MO, USA), vancomycin 0.5 g/L (SBR00001, Sigma-Aldrich, Saint Louis, MO, USA), and kanamycin 0.5 g/L (Beyotine, Shanghai, China) was prepared and applied in drinking water.

### 4.4. Sample Collection

Before sampling, body weight was recorded on days 7 and 14 of the experiment. Subsequently, on days 7 and 14, six broilers from each treatment were randomly selected, weighed and euthanized by cervical dislocation. Blood was taken from the wing vein and centrifuged (3000× *g*, 10 min) at 4 °C, and then the serum was harvested and stored at −20 °C until analysis.

The weight and length of the small intestine (duodenum, jejunum, and ileum) were recorded and presented as a percentage of live BW (cm/kg) based on a previous study conducted by Mahdzvi and Torki (2009) [78]. The weight of the liver, spleen, and bursa of Fabricius was measured and expressed as a percentage of live BW (g kg). Subsequently, the proximal ends of the ileum were snap-frozen in liquid nitrogen and stored at −80 °C for mRNA analysis. Also, ~2 cm-long duodenal, jejunal, and ileal samples, taken midway between the endpoint of the duodenal loop and Meckel’s diverticulum, were collected, flushed with 10% neutral buffered formalin solution, and fixed overnight in 10% neutral buffered formalin solution for histological examination. The ileal and cecal contents were aseptically collected and frozen immediately for microbiome analysis.

### 4.5. Intestinal Histomorphological Analysis

Gut histomorphology (villus height (VH) and crypt depth (CD)) and goblet cell analysis were performed as previously described [79]. The density of goblet cells and goblet cells producing acidic sialylated mucins in the intestine was measured through Alcian blue (AB) and periodic acid–Schiff (PAS) staining. The number of goblet cells in each villus was measured, and the density of goblet cells was calculated as the number of goblet cells per micrometer of villus height.

### 4.6. Assay for Immunoglobulins, Cytokines and LPS in the Serum, and DAO and MPO Activity in the Ileum

Levels of immunoglobulin (immunoglobulin G (*IgG*) and *IgA*), cytokines, including interleukin (*IL-2*, *IL-4*, *IL-6*, tumor necrosis factor-alpha (*TNF-α*), and interferon gamma (*IFN-γ*) and lipopolysaccharide (LPS) in the serum were quantified using enzyme-linked immunosorbent assay (ELISA) kits specifically for chickens (Shanghai Enzyme-linked Biotechnology Co., Ltd.) following the manufacturer’s instructions. Diamine oxidase (DAO) and myeloperoxidase (MPO) activity in the ileum mucosa were measured using commercial kits according to the provider’s instructions (Nanjing Jiancheng Institute of Biological Engineering, Nanjing, China).

### 4.7. Quantitative Real-Time PCR

Total RNA was extracted from the ileum tissue (50–100 mg) using Trizol agent (Tiangen Biotech Co., Ltd., Beijing, China) following the manufacturer’s instructions. A NanoDrop 2000 spectrophotometer (Thermo Fisher Scientific, Waltham, MA, USA) was used to determine the concentration and purity of total RNA. Subsequently, cDNA synthesis was carried out using a PrimeScriptTM RT reagent kit with a gDNA Eraser (perfect real time) kit (Takara Biotechnology Co., Ltd., Beijing, China). Quantitative real-time PCR (qRT-PCR) assays were conducted on an Applied Biosystems 7500 Fast Real-Time PCR system using an SYBR Premix Ex-Taq diagnostic kit (Takara Biotechnology Co., Ltd., Beijing, China), and each sample was analyzed in duplicate. The β-actin gene served as the reference gene to standardize the mRNA levels for the target genes, which included *OCLN*, *ZO-1*, *MUC-2*, *CLDN-1*, *FABP-2*, *IL-1β*, *IL-6*, *TNF-α*, *IFN-γ*, and *IL-8*. Primer sequences for this analysis are provided in Appendix A. The expression level of each target gene was normalized by the comparative cycle threshold (CT) 2−ΔΔCT method [80].

### 4.8. Short-Chain Fatty Acid Determination in Cecal Contents

The frozen cecal digesta samples (100 mg) from each replicate were dissolved and homogenized in 1.5 mL of ice-cold sterile ultrapure water and centrifuged at 12,000× *g* at 4 °C for 10 min. Subsequently, 1 mL of the supernatant was carefully diluted with 0.2 mL of 25% (*w*/*v*) metaphosphoric acid solution, which also included crotonic acid. This mixture was then incubated at −20 °C for 24 h and again centrifuged at 10,000× *g* for 10 min at 4 °C to eliminate the protein precipitates. The resulting solution was filtered using a 0.22 μm syringe filter for purification. The analysis of short-chain fatty acids (SCFAs) was carried out using a Shimadzu GC-2014 ATF gas chromatograph equipped with a capillary column (30 m × 0.25 mm × 0.5 μm). The N2 was used for carrier gas (12.5 Mpa, 18,562 mL/min). The temperatures of the injector and detector were maintained at 180 °C, while the column temperature was gradually increased from 80 °C to 170 °C at a rate of 5 °C/min. The SCFA concentrations were described in milligrams per kilogram of digesta.

### 4.9. Microbial DNA Extraction and 16S rRNA Sequencing and Analysis

Microbial genomic DNA was isolated using roughly 250 mg of cecal digesta samples collected from all experimental groups, employing the EZNA^®^ Soil DNA Kit (Omega Bio-Tek, Norcross, GA, USA) following the manufacturer’s instructions. The total DNA concentration and its purity were assessed using the NanoDrop 2000 spectrophotometer (Thermo Scientific, Waltham, MA, USA), while the DNA integrity was evaluated through 1% agarose gel electrophoresis, applying a voltage of 5 V/cm for 20 min. The V3–V4 regions of the bacterial 16S rDNA sequences were amplified with the 338F (5′-ACTCCTACGGGAGGCAGCAG-3′) and 806R (5′-GGACTACHVGGGTWTCTAAT-3′) primer pair following a previously described methodology [81]. The PCR products were then purified using the AxyPrep-DNA gel extraction kit (Axygen, Union City, CA, USA), quantified, homogenized, and used for constructing the MiSeq library. This library was sequenced on the Illumina MiSeq platform (Illumina, Santa Clara, CA, USA) using a MiSeq reagent kit from Shanghai Personal Biotechnology Co., Ltd., Shanghai, China. The raw pair-end sequences underwent demultiplexing and quality filtering through the quantitative insights into microbial ecology (QIIME, V1.17) software [82]. Effective reads were grouped into operational taxonomic units (OTUs) with 97% similarity. OTU classification at various taxonomic levels utilized the Greengenes database. Analysis of rarefaction curves and α diversity (including the Chao 1 index, Simpson index, Ace index, and Shannon index) was performed using QIIME software [83]. Additionally, β diversity was assessed through principal coordinate analysis (PCoA) and partial least squares discriminant analysis, with results visualized using the “vegan” and “ggplot2” R packages (R-V3.4.4). Differences in microbial communities across groups were examined using ANOSIM implemented in the “vegan” R package [84]. The impact of bacterial abundance differences from the phylum to genus level among groups was determined using linear discriminant analysis effect size (LEfSe) analysis (LDA score > 4.0, *p* < 0.05). Statistical analysis of non-parametric factors was conducted using the Kruskal–Wallis rank-sum test [85].

### 4.10. Data Analysis

One-way analysis of variance (ANOVA) followed by Duncan’s multiple comparison test was performed using SPSS (version 21.0 from Chicago, IL, USA) to analyze the growth performance, intestinal morphology, gene expression, and SCFA contents, with *p* < 0.05 considered significant and 0.05 ≤ *p* < 0.10 considered a trend. The results are expressed as means and pooled SEM. Correlations were analyzed using Spearman’s correlation with the heatmap package, with a *p*-value less than 0.05 considered significant.

## 5. Conclusions

Drinking water supplementation with an antibiotic cocktail in early chickens significantly altered ileal and cecal intestinal microbiota composition and impaired intestinal microbiota homeostasis, as evidenced by increasing α diversity of gut microbiota, inhibiting the growth of potential beneficial strains such as *Lactobacillus* and *Bacillus*, while promoting the proliferation of antibiotic-resistant pathogens like *Enterococcus*. This alteration resulted in suppressing SCFA generation, which eventually led to delayed intestinal morphological development, disrupted intestinal barrier function and perturbed immune response in early chickens. Furthermore, prolonged (long-term) use of the antibiotic cocktail was more harmful to gut health than short-term use. This study provides a reliable lower-bacteria chicken model for further investigating the function of certain beneficial bacteria in the gut by fecal microbiota transplantation into germfree or antibiotic-treated chickens.

## Figures and Tables

**Figure 1 antibiotics-13-00413-f001:**
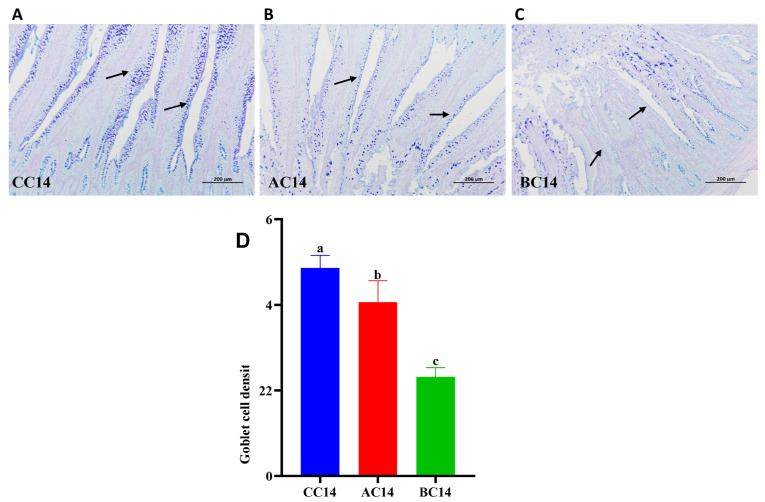
Effects of antibiotic cocktail (ABX) treatment on the intestinal mucus layer and ultrastructure. Control group (CC) showing healthy mucus layer with the mucin-secreting cells where basal feed with boiled water only was supplied (**A**), while AC and BC were fed basal diet with ABX for 7 days and ABX for 14 days, respectively, in the drinking water at a rate of 2.5 g/L. The AC group showed a slight effect of ABX on the ultrastructure of mucin and mucin-secreting cells (**B**), and the BC group showed significant effects of 14 days’ ABX treatment on both the ultrastructure of mucin and mucin-secreting cells (**C**). The mucin-secreting cells were successively reduced with extended application of ABX (**D**).

**Figure 2 antibiotics-13-00413-f002:**
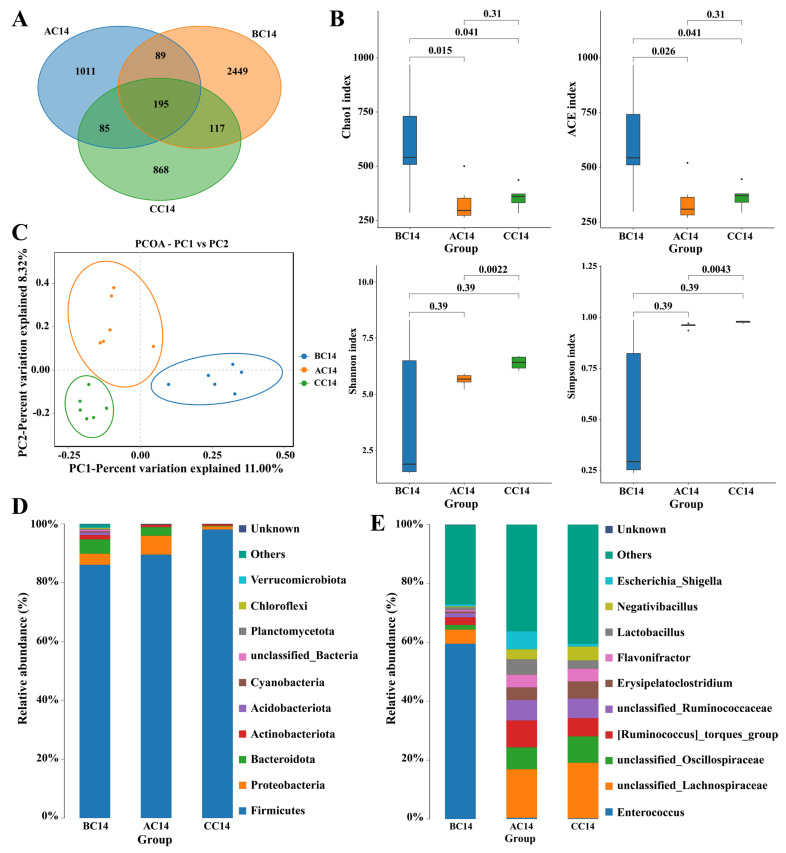
Antibiotic cocktail (ABX) effects on the composition and function of the cecal microbiota after 14 days of sampling. (**A**) Venn diagram showing operational taxonomic units (OTUs) shared by the three groups based on 97% sequence similarity. (**B**) The α diversity as measured by the CE index (i.e., i. ACE index, ii. Chao index, iii. Shannon index, iv. Simpson index) among all three groups. (**C**) The β diversity measured different microbial communities among all groups (CC14, AC14, and BC14) using principal component analysis (PCoA). The relative abundance at phylum **(D**) and genus level (**E**) is degerming the microbial communities.

**Figure 3 antibiotics-13-00413-f003:**
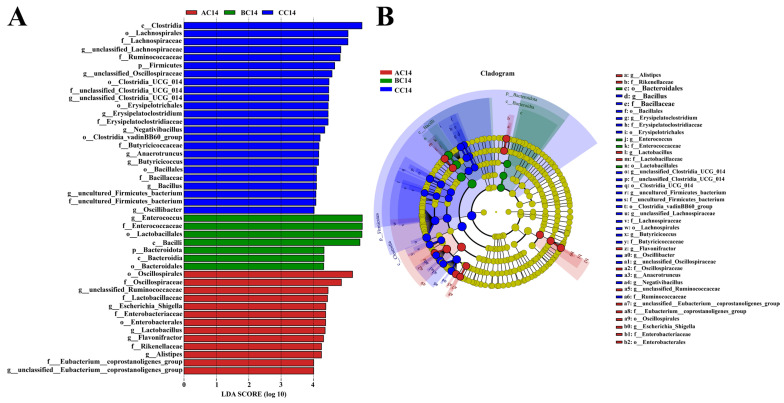
Antibiotic cocktail (ABX) effects on the differential bacterial taxa in the cecal microbiota. (**A**) LEfSe analysis of intestinal microbiota composition after ABX treatment based on LDA score (log 10). (**B**) Cladogram exhibiting differential bacteria in the control group (CC14) and a significant difference in the antibiotic groups (AC14 and BC14).

**Figure 4 antibiotics-13-00413-f004:**
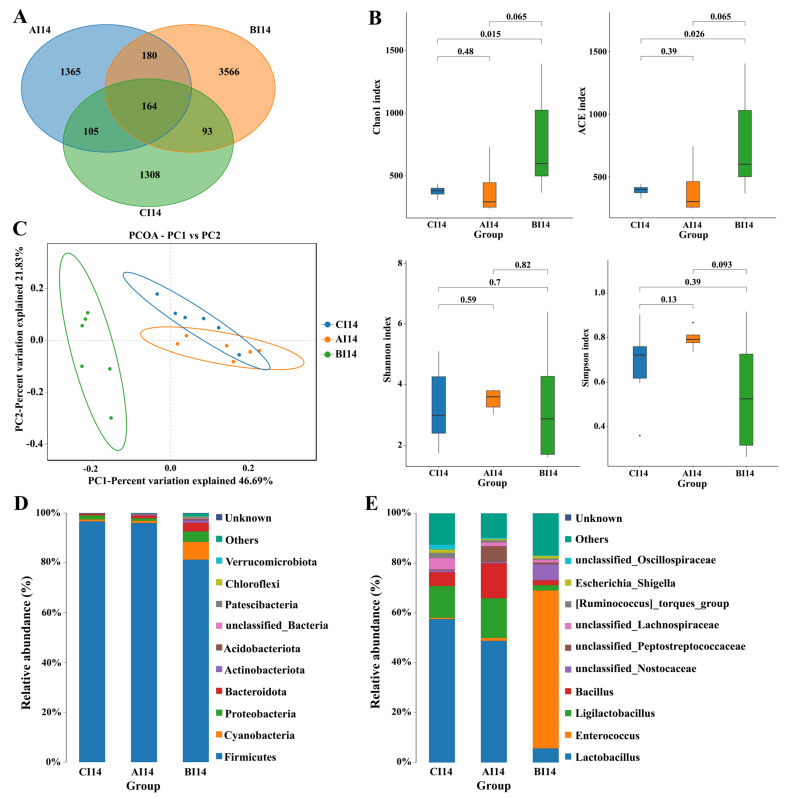
Antibiotic cocktail (ABX) effects on ileal microbiota, including their composition and function, after 14 days of sampling. (**A**) Venn diagram presenting operational taxonomic units (OTUs) shared by the three groups based on 97% sequence similarity. (**B**) The α diversity from the CE index (ACE index, Chao index, Shannon index, and Simpson index) among all three groups. (**C**) The β diversity for different ileum microbial communities among all groups (CC14, AC14, and BC14) using principal component analysis (PCoA). The relative abundance at phylum (**D**) and genus level (**E**), indicating the microbial communities’ top 10 species at the phylum and genus level.

**Figure 5 antibiotics-13-00413-f005:**
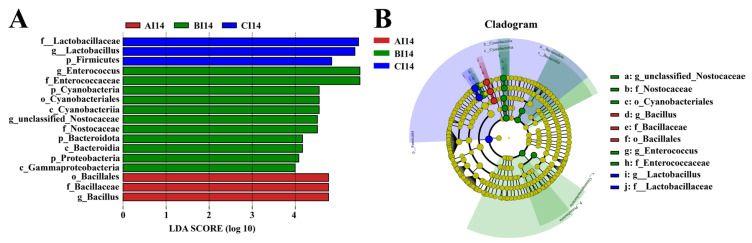
Antibiotic cocktail (ABX) effects on the different taxa of the cecal microbiota. (**A**) LEfSe analysis of ileal microbiota composition after 14 days of antibiotic cocktail treatment based on LDA score (log 10). (**B**) Cladogram demonstrating differential bacteria in the control group (CI14), with a significant difference in the antibiotic groups (AI14 and BI14).

**Figure 6 antibiotics-13-00413-f006:**
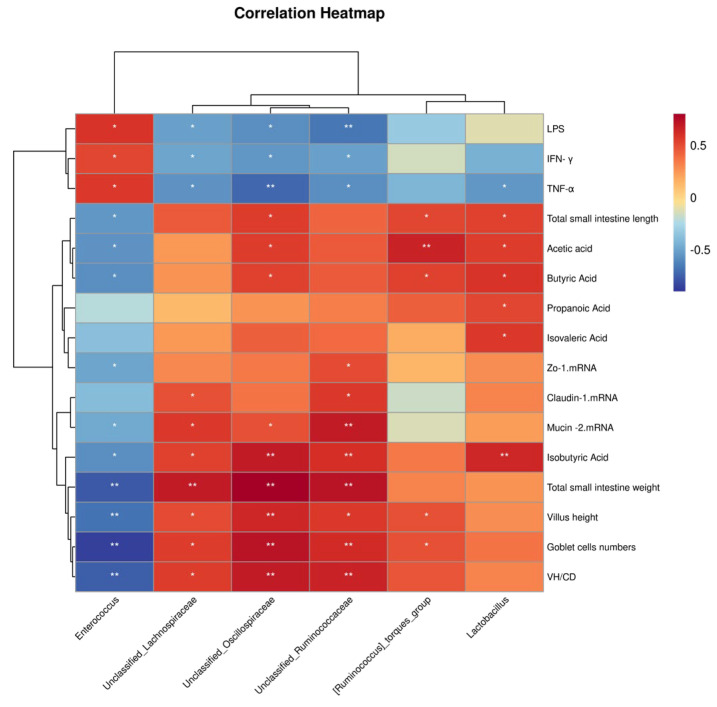
Correlation analysis of the cecal microbiota. Spearman correlation analysis between phenotypic variables and the relative abundance of microbial communities with significant differences (genus level, *n* = 6/group). The color and dot size represent correlation coefficients within rows. * *p* < 0.05, ** *p* < 0.01.

**Table 1 antibiotics-13-00413-t001:** Effects of antibiotic cocktail (ABX) treatment on broiler weight gain, organ index, and small intestinal length and weight (*n* = 6).

Items	CC14	AC14	BC14	SEM	*p*-Values
**Body weight gain (g)**					
Day 7	173.10	189.91	193.41	4.22	0.148
Day 14	472.53	451.80	467.08	7.34	0.518
**Organ index (g/kg)**					
Liver index	27.85	29.29	25.37	0.74	0.145
Spleen index	0.71	0.81	0.85	0.37	0.329
Bursa index	1.86	1.96	2.10	0.117	0.783
**Intestinal length (cm/kg)**					
Duodenum length	7.73 ^a^	4.48 ^c^	6.05 ^b^	0.411	0.001
Jejunum length	14.32	12.42	14.83	0.676	0.328
Ileum length	10.75	9.70	9.72	0.425	0.546
Total small intestinal length	32.79 ^a^	26.60 ^b^	30.60 ^ab^	1.191	0.092
**Small intestinal weight (g/kg)**					
Duodenum weight	10.250 ^a^	7.135 ^b^	5.472 ^b^	0.646	<0.001
Jejunum weight	16.647 ^a^	14.973 ^a^	12.278 ^b^	0.553	<0.001
Ileum weight	12.467 ^a^	9.937 ^b^	9.007 ^b^	0.446	<0.001
Total small intestinal weight	39.363 ^a^	32.048 ^b^	26.753 ^c^	1.509	<0.001

SEM, standard error of the mean; CC, control group fed basal diet with non-antibiotic boiled water only, while AC and BC were fed basal diet with ABX for 7 days and ABX for 14 days, respectively, in the drinking water at a rate of 2.5 g/L. ^a,b,c^ Values with different superscripts differ significantly (*p* < 0.05).

**Table 2 antibiotics-13-00413-t002:** Effects of antibiotic cocktail (ABX) treatment on the morphological structure of the small intestine in early broilers (*n* = 6).

Items	CC14	AC14	BC14	SEM	*p*-Values
**Duodenum**					
Villus height, μm	1441.33 ^a^	1239.16 ^b^	1043.00 ^c^	49.53	<0.001
Crypt depth, μm	235.50 ^a^	269.50 ^b^	222.50 ^a^	5.79	<0.001
VH/CD	6.16 ^a^	4.61 ^b^	4.59 ^b^	0.23	<0.002
**Jejunum**					
Villus height, μm	955.16 ^a^	767.83 ^b^	642.83 ^c^	32.72	<0.001
Crypt depth, μm	127.66 ^b^	196.00 ^a^	206.33 ^a^	9.53	<0.001
VH/CD	7.55 ^a^	4.00 ^b^	3.11 ^c^	0.49	<0.001
**Ileum**					
Villus height, μm	662.66 ^a^	562.00 ^b^	512.22 ^c^	16.54	<0.001
Crypt depth, μm	95.00 ^b^	113.16 ^b^	150.01 ^a^	6.55	<0.001
VH/CD	7.11 ^a^	4.97 ^b^	3.477 ^c^	0.398	<0.001

SEM, standard error of the mean; CC, control group given basal feed with non-antibiotic boiled water only, while AC and BC were fed basal diet with ABX for 7 days and ABX for 14 days, respectively, in the drinking water at a rate of 2.5 g/L. ^a,b,c^ Values with different superscripts differ significantly (*p*< 0.05).

**Table 3 antibiotics-13-00413-t003:** Effects of antibiotic cocktail (ABX) treatment on short-chain fatty acid (SCFA) profiles of cecal contents in broilers (*n* = 6).

Items (μmol/g)	CC14	AC14	BC14	SEM	*p*-Values
Acetic acid	41.42 ^b^	52.15 ^a^	4.34 ^c^	5.165	<0.001
Propanoic acid	1.48 ^b^	3.11 ^a^	1.31 ^b^	0.220	<0.001
Isobutyric acid	0.30 ^a^	0.35 ^a^	0.12 ^b^	0.034	<0.006
Butyric acid	3.84 ^b^	6.60 ^a^	0.46 ^c^	0.677	<0.001
Isovaleric acid	0.293 ^ab^	0.527 ^a^	0.219 ^b^	0.055	0.004
Valeric acid	0.18	0.15	0.16	0.011	0.363

SEM, standard error of the mean; CC, control group fed basal diet with boiled water only, while AC and BC were fed basal diet with ABX for 7 days and ABX for 14 days, respectively, in the drinking water at a rate of 2.5 g/L. ^a,b,c^ Values with different superscripts differ significantly (*p* < 0.05).

**Table 4 antibiotics-13-00413-t004:** Effects of antibiotic cocktail (ABX) treatment on systemic and intestinal mucosal immune functions in broilers (*n* = 6).

Items	CC14	AC14	BC14	SEM	*p*-Values
**Systemic immune responses**					
IFN-γ (pg/mL)	52.16 ^b^	64.49 ^ab^	73.65 ^a^	3.53	0.033
TNF-α (pg/mL)	33.91 ^b^	37.83 ^b^	45.81 ^a^	1.41	0.001
IL-2 (pg/mL)	102.76	107.76	133.76	9.28	0.377
IgG (μg/mL)	1428.27 ^b^	1744.46 ^a^	1528.26 ^a^	42.70	0.011
IgA (μg/mL)	183.92 ^a^	152.63 ^b^	167.63 ^b^	5.31	0.004
**Ileal immune responses**					
IL-1β	1.00 ^a^	0.39 ^ab^	0.23 ^b^	0.138	0.047
IL-6	1.00	0.43	0.79	0.223	0.641
IL-8	1.00 ^a^	0.39 ^ab^	0.31 ^b^	0.138	0.057
IFN-r	1.00 ^a^	0.86 ^ab^	0.37 ^b^	0.118	0.082
TNF-a	1.00 ^a^	0.51 ^b^	0.28 ^b^	0.118	0.016

SEM, standard error of the mean; CC, control group fed basal diet with boiled water only, while AC and BC were fed basal diet with ABX for 7 days and ABX for 14 days, respectively, in the drinking water at a rate of 2.5 g/L. ^a,b^ Values with different superscripts differ significantly (*p* < 0.05).

**Table 5 antibiotics-13-00413-t005:** Effects of antibiotic cocktail (ABX) treatment on intestinal permeability and barrier-related gene expression in the ileum of broilers (*n* = 6).

Items	CC14	AC14	BC14	SEM	*p*-Value
**Intestinal permeability**					
LPS (EU/L)	47.48 ^b^	85.29 ^a^	114.50 ^a^	8.87	0.002
DAO (ng/mL)	10.01 ^b^	9.11 ^b^	30.50 ^a^	3.35	0.006
MPO (pg/mL)	692.83	701.33	741.66	9.10	0.057
**Barrier-related gene expression**					
ZO-1	1.00 ^ab^	0.26 ^ab^	0.10 ^b^	0.163	0.041
Occludin	1.00	0.21	0.09	0.234	0.242
Mucin 2	1.00 ^a^	0.30 ^ab^	0.07 ^b^	0.166	0.030
FABP-2	1.00	1.27	0.46	0.299	0.558
Claudin 1	1.00 ^a^	1.34 ^a^	0.34 ^b^	0.201	0.042

SEM, standard error of the mean; CC, control group fed basal diet with boiled water only, while AC and BC were fed basal diet with ABX for 7 days and ABX for 14 days, respectively, in the drinking water at a rate of 2.5 g/L, ^a,b^ Values with different superscripts differ significantly (*p* < 0.05).

## Data Availability

The original contributions presented in the study are publicly available in the NCBI repository (accession number PRJNA1083794).

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
