# Peer review of "Antibiotic Cocktail Effects on Intestinal Microbial Community, Barrier Function, and Immune Function in Early Broiler Chickens"

_antibiotics, 2024, doi:10.3390/antibiotics13050413_

Round 1

Reviewer 1 Report

Comments and Suggestions for Authors

This study describes the effects of antibiotics cocktails on intestinal microbial compositions and their characteristics, including body length, weight, morphological structure, fatty acid composition, immune responses, gene expression, and microbial composition. However, this needs more discussion on the distribution of antibiotic resistant bacteria when treated with antibiotic cocktails.

1.     Fatty acid compositions will be influenced by the presence of intestinal bacterial. Their viability varies with the degree of antibiotic resistance. Then, the antibiotic resistance profiles should be tested for all survival bacteria in comparison of control group.

2.     The bioconversion ability of viable intestinal bacteria should be related the amount of fatty acid compositions and other substances.

3.     All properties should be compared with the antibiotic resistance profile of viable intestinal bacteria. This might be a suitable comparison and provide better information.   

Comments on the Quality of English Language

Moderate editing of English language required.

Reviewer 2 Report

Comments and Suggestions for Authors

The authors present data on the effects of drinking water application for 7 and 14 days of an antibiotic cocktail on various immune parameters, microbiologic examinations, and gut histology.

A few comments/concerns:

The microbial analysis (microbiota changes and populations data) figures should be larger for ease of observation. All Table and Figure legends should include more information as to the assays portrayed in them.

It would be interesting to see the individual effects of each antibiotic on immune and microbiologic parameters measured here. Not necessary for the studies presented.

The experimental design had 30 birds/group with 6 birds/group used in the analysis. Was this study repeated or was it conducted only one time?

Otherwise, scientifically sound and interesting data shown.

Comments on the Quality of English Language

There are some instances of English/grammar mistakes, but not overwhelming.

Round 2

Reviewer 1 Report

Comments and Suggestions for Authors

It has been well revised.

Comments on the Quality of English Language

Moderate editing of English language required